# ENVIRONMENT-AWARE ON-MANIFOLD 3D TEXTURE CAMOUFLAGE FOR PHYSICAL ATTACKS ON VEHICLE DETECTORS

## ABSTRACT

We study full-coverage, printable 3D camouflage attacks on vehicle detectors. Our pipeline decouples photorealism from attackability by combining a closed-form *Intrinsic Appearance Transfer* (IAT) module with an on-manifold StyleGAN texture prior under Expectation-over-Transformations (EOT) focused on camera and environment. IAT carries exposure/white balance/tone and veiling from a reference frame to the render via per-pixel affine carriers and is training-free at test time; adversarial textures are optimized only through early StyleGAN layers to preserve material plausibility. On a scene-controlled CARLA corpus spanning 22 weather/time presets, 8 azimuths, 9 elevations, 6 distances, and 3 locations, our method—optimized white-box on YOLOv3 and evaluated black-box on Faster R-CNN, RetinaNet, RTMDet, and DINO—reduces AP@0.5 from 0.75 to 0.11 on YOLOv3 ($-85.8\%$), with corresponding drops to 0.13 ($-82.5\%$) on Faster R-CNN, 0.22 ($-68.7\%$) on RetinaNet, 0.26 ($-67.1\%$) on RTMDet, and 0.59 ($-31.7\%$) on DINO. Averaged over detectors, AP@0.5 decreases from 0.7538 to 0.2863 ($\approx 62\%$). Ablations show that (i) sRGB-domain affine fits excel on unseen *colors*, while linear-RGB fits excel on unseen *textures*; and (ii) cross-color U-Net training with a content loss yields the best perceptual fidelity among learned baselines. Overall, a simple, differentiable IAT combined with a layer-restricted generative prior offers a practical path to robust, photorealistic 3D camouflage that transfers across models and conditions.

## 1 INTRODUCTION

Deep neural networks (DNNs) have transformed perception in safety–critical domains such as autonomous vehicles (AVs) (Bojarski et al., 2016; Chen et al., 2017; Kuutti et al., 2020). Yet modern vehicle detectors remain vulnerable to *physical* adversarial attacks, where full-coverage textures applied to the object itself suppress detections or induce misclassification (Goodfellow et al., 2014; Carlini & Wagner, 2017; Zhang et al., 2019; Huang et al., 2020; Duan et al., 2020). Unlike purely digital perturbations, physical camouflages must survive the entire digital-to-physical (D2P) pipeline—changes in viewpoint and distance, cast shadows, view-dependent reflections, weather, and sensor characteristics—while remaining printable and fabricable.

Prior full-coverage pipelines can be grouped into three families. *World-aligned* (e.g., triplanar/projection) methods optimize textures in world coordinates to encourage universality and cross-instance transfer, but are sensitive to deployment misalignment and pose/distance shifts, creating train–deploy gaps (Suryanto et al., 2023; 2022). *UV-map* methods instead optimize a single full-body texture in the vehicle's UV space via differentiable rendering, enabling precise deployment but historically lacking robust modeling of scene photometry (illumination, shadows, reflections, weather) and end-to-end UV optimization across weather/time shifts (Wang et al., 2021; 2022; Zhou et al., 2025; Lyu et al., 2024). A third, *neural-field* family leverages differentiable volumetric/splatting renderers (e.g., 3D Gaussian Splatting) to obtain more photorealistic, multi-view-consistent images without relying solely on mesh+UV assumptions (Lou et al., 2025). We adopt a mesh+UV rasterizer (e.g., PyTorch3D/NMR (Ravi et al., 2020; Kato et al., 2018)) because it provides a stable, printable UV parameterization with deterministic barycentric sampling and direct gradients to texels—cleanly separating reflectance from illumination—whereas neural splatting is

often trained per-scene and does not yield a unified UV layout for a single physical wrap (Lou et al., 2025).

Despite this progress, a central limitation persists: insufficient *photorealism* in the D2P loop. UV-based pipelines frequently omit scene-consistent carriers—cast shadows, view-dependent reflections, veiling/haze, and weather—which hurts realism and attack stability; even environment modules built with encoder–decoder networks only partially address these effects (Zhou et al., 2025). At the same time, unconstrained pixel-level optimization often produces high-frequency, conspicuous patterns that overfit a source detector and limit black-box transfer (Wang et al., 2024; Zhang et al., 2023). Orthogonal to the mapping choice, objective-driven works use diffusion or contrastive learning to encourage naturalness and transfer (Lyu et al., 2024; Zhang et al., 2025). We also note that Wang et al. (2025) focus on transferability using a differentiable renderer with attention-dispersion and enhanced training strategies—not on world-aligned mappings.

We decouple *environment appearance transfer* from *adversarial texture design* and confine optimization to an *on-manifold* generative prior. First, we introduce a lightweight *Intrinsic Appearance Transfer* (Intrinsic Appearance Transfer (IAT)): a closed-form affine radiometry that carries exposure/white-balance/tone and veiling from the input frame to the rendered vehicle. Unlike U-Net/DenseNet environment modules, our affine formulation is training-free at test time, fully differentiable, and—crucially—generalizes better to *unseen* textures and scenes. Second, we parameterize the UV texture with a *StyleGAN prior* (Karras et al., 2021) and restrict updates to physically meaningful, early-layer edits. This avoids per-pixel "micro-nudges," produces material-plausible patterns, and improves physical transfer. Third, we shape the attack loss to operate at the detector level (objectness/classification/localization aggregated across views and distances), which empirically yields stronger black-box transfer to held-out detectors. Finally, we train and evaluate on a scene-controlled CARLA corpus spanning maps, weather, time-of-day, and poses, including paired cross-reflectance renders (same pose/scene, different base paints) to supervise illumination transfer.

**Contributions.**

- A simple, closed-form and differentiable **IAT** that *separates illumination from reflectance* via per-pixel affine transfer, outperforming U-Net/DenseNet environment modules in generalization to unseen colors, textures, and scenes.

- A **StyleGAN-prior** UV-texture attack with *layer-restricted* updates, yielding natural, printable wraps that improve black-box transfer compared with noise-initialized pixel optimization.

- A practical, end-to-end **training pipeline** (Blender UV remap, PyTorch3D renderer, focused EOT over camera and environment) that achieves stronger mAP drops on a white-box source and better cross-model transfer than texture-only or heavy environment-network baselines—e.g., DAS, FCA, ACTIVE, RAUCA, CNCA, PhyCamo (Wang et al., 2021; 2022; Suryanto et al., 2023; Zhou et al., 2025; Lyu et al., 2024; Zhang et al., 2025)—while remaining easy to implement. Transfer is measured on *Faster R-CNN*, *RetinaNet*, *RTMDet*, and *DINO* (Ren, 2015; Lin, 2017; Lyu et al., 2022; Zhang et al., 2022).

**Threat model.** We assume white-box access to one source detector during optimization and evaluate transfer to held-out detectors at test time. The attacker may modify only the vehicle's UV texture via a StyleGAN latent; geometry, camera intrinsics/extrinsics, and scene layout are fixed. Expectation-over-Transformations covers camera pose and CARLA environment (weather/time/map). The default goal is untargeted suppression (objectness/classification/localization), with optional targeted variants. Physical constraints include latent-norm bounds and an optional printability prior; no digital test-time tampering is allowed. Transfer is measured on Faster R-CNN, RetinaNet, RTMDet, and DINO (Ren, 2015; Lin, 2017; Lyu et al., 2022; Zhang et al., 2022).

## 2 BACKGROUND AND RELATED WORK

Adversarial vehicle camouflage spans *physical* attacks, differentiable 3D rendering, and priors for *naturalness* and *transferability*. We group prior art into (i) full-coverage camouflage, (ii) natural/stealthy generation, (iii) transferability strategies, and (iv) rendering & EOT. Our design departs via an illumination-consistent appearance module (IAT), a StyleGAN prior with layer restriction, Blender UV remapping, and a camera/environment-focused EOT.

**Full-coverage camouflage.** Physical attacks progressed from EOT-robust digital examples and localized patches (Athalye et al., 2018; Liu et al., 2018; Thys et al., 2019) to full-vehicle wraps via proxy/differentiable optimization (Zhang et al., 2019). Two mappings dominate: *world-aligned* (triplanar/projection) favor universality but break under deployment misalignment (Suryanto et al., 2023; 2022); *UV-map* methods enable precise, printable wraps but historically under-model scene photometry and end-to-end UV optimization across weather/time (Wang et al., 2021; 2022; Zhou et al., 2024; 2025; Lyu et al., 2024). Neural-field renderers (e.g., 3D Gaussian Splatting) offer high-fidelity multi-view synthesis but are usually trained per scene and lack a unified printable UV layout (Lou et al., 2025). FCA/ACTIVE strengthen multi-view robustness (Wang et al., 2022; Suryanto et al., 2023); RAUCA adds an environment feature extractor for cross-weather gains (Zhou et al., 2024; 2025); CamoEnv aligns object/environment with an implicit color module to improve consistency and black-box transfer (Zhu et al., 2025).

**Natural and stealthy camouflage.** Generative priors aim for on-manifold, less conspicuous textures: diffusion-guided CNCA produces customizable natural wraps (Lyu et al., 2024), while Phy-Camo couples diffusion augmentation with a contrastive objective to boost multi-view robustness and transfer (Zhang et al., 2025). We pursue naturalness via a *StyleGAN prior* with *layer-restricted* edits, avoiding heavy diffusion at optimization time (Karras et al., 2021).

**Transferability and universality.** Cross-model transfer remains hard. Attention suppression/redistribution and training recipes improve transfer for physical attacks (Wang et al., 2021; Zhang et al., 2023; Wang et al., 2024); "highly transferable" camouflage combines attention dispersion with enhanced training atop a differentiable renderer (Wang et al., 2025); gradient calibration/regularization stabilizes view sensitivity (Liang et al., 2025). Our coupling of a StyleGAN prior with *illumination-consistent* IAT reduces dependence on paint/lighting statistics and supports cross-condition, cross-detector generalization.

**Rendering, compositing, and EOT.** Differentiable rasterization (Kato et al., 2018; Ravi et al., 2020) passes detector gradients to texels; EOT samples pose, distance, and environment to promote robustness (Athalye et al., 2018). We adopt a mesh+UV rasterizer (PyTorch3D) for deterministic barycentric sampling and direct texel gradients, then *explicitly* inject scene photometry via a closed-form IAT before compositing over CARLA (Dosovitskiy et al., 2017). Neural splatting improves fidelity but typically bakes lighting from training imagery and does not yield a single printable UV layout (Lou et al., 2025).

**GAN-based adversarial generation.** Generators have been trained to output adversarial perturbations (e.g., GAP) and to blend adversarial signals with plausible styles via conditional GANs/perceptual losses (Poursaeed et al., 2018; Isola et al., 2017; Johnson et al., 2016; Gatys et al., 2016). Most, however, target the *digital* image domain or patches and do not provide a *single, printable UV texture* or explicit handling of illumination (shadows, view-dependent reflections, weather). We instead *constrain* optimization with a pre-trained StyleGAN prior and early-layer restriction, while a closed-form IAT enforces scene-consistent photometry (Karras et al., 2021).

## 3 METHOD

### 3.1 FRAMEWORK OVERVIEW

We couple a differentiable renderer PyTorch3D, an illumination-consistent image-formation module (Intrinsic Appearance Transfer (IAT)), and a generative texture prior (Fig. 1). From a CARLA frame $I_{\text{in}}$ with mask $M_{\text{fg}} \in \{0, 1\}^{H \times W}$ and camera $\phi_{\text{cam}}$, we form

$$B_{\text{bg}} = I_{\text{in}} \odot (1 - M_{\text{fg}}), \qquad I_{\text{veh}} = I_{\text{in}} \odot M_{\text{fg}}.$$

Let $e$ denote the environment preset (weather, time-of-day, map); we write $\phi = (\phi_{\text{cam}}, e)$.

**Generative texture & UV coating.** We sample a StyleGAN latent $\mathbf{z}$ (initialized at $\mathbf{z}_0$) and decode a $256 \times 256$ texture $P = G(\mathbf{z})$. With our Blender-remapped UV atlas, PyTorch3D rasterizes the textured mesh (native UV sampling) under $\phi_{\text{cam}}$ to produce $I_r$.

**Intrinsic Appearance Transfer (IAT).** Intrinsic Appearance Transfer (IAT) transfers scene illumination/veiling from the input vehicle region onto the render via per-pixel, channel-wise carriers

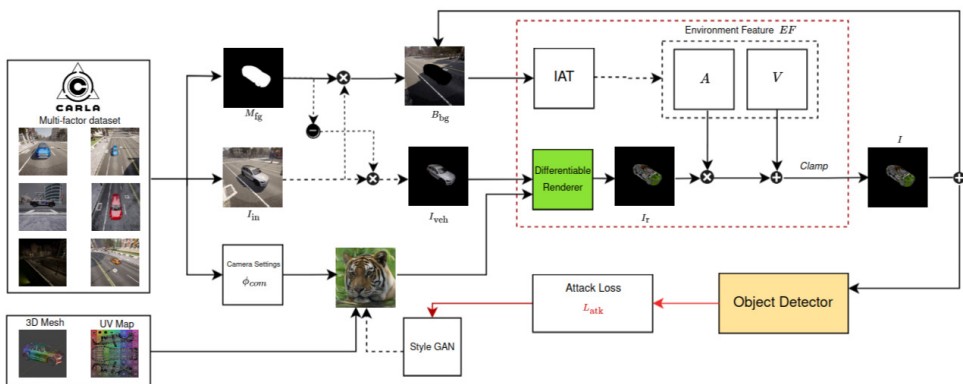

Figure 1: **Framework.** A StyleGAN-prior UV texture is rendered via PyTorch3D on a Blender-remapped atlas; Intrinsic Appearance Transfer (IAT) (closed-form affine) transfers illumination/veiling from the input frame; detector losses backpropagate through renderer $\rightarrow$ IAT $\rightarrow$ early StyleGAN layers under EOT over camera and environment.

$(A, V)$ (RGB gains and offsets),

$$\tilde{I} = \mathrm{IAT}(I_r;\, I_{\mathrm{veh}}) = A \odot I_r + V, \qquad I = M_{\mathrm{fg}} \odot \tilde{I} + (1 - M_{\mathrm{fg}}) \odot B_{\mathrm{bg}}.$$

During attacks, Intrinsic Appearance Transfer (IAT) is *pretrained and frozen* but *differentiable*, so gradients backpropagate through PyTorch3D, UV sampling, and Intrinsic Appearance Transfer (IAT) into the StyleGAN latent. We use the *closed-form affine* IAT by default; the U-Net variant appears only in ablations.

**Attack objective.** Over an Expectation-over-Transformations (EOT) sampler, we draw $(\phi_{\mathrm{cam}}, e) \sim \mathcal{T}$ and optimize only the StyleGAN latent (all other parameters fixed):

$$\min_{\mathbf{W}}\ \mathbb{E}_{(\phi_{\mathrm{cam}}, e) \sim \mathcal{T}}\big[\mathcal{L}_{\mathrm{attack}}(\mathcal{D}(I))\big] + \lambda_w\, \mathcal{R}_w,$$

where $\mathcal{D}$ is the detector and $\mathcal{R}_w$ is a latent-space regularizer acting only on the trainable (coarse) styles (Sec. 3.4/3.5). We *do not* use pixel TV by default (rationale in Sec. 3.4).

## 3.2 DIFFERENTIABLE 3D PIPELINE

We use a scene-controlled CARLA corpus in which, for each fixed pose, we generate *paired* frames—identical geometry/camera with different base appearances (17 colors/textures)—to supervise cross-reflectance transfer; for attacks we vary map locations and standard weather/time presets, since environment effects (cast-shadow geometry, interreflections, veiling) are location-dependent. To enable seamless coating, we re-unwrap the vehicle in **Blender** to merge large panels into connected UV islands, equalize texel density, and align seams with low-salience edges. Rendering uses **PyTorch3D** with native UV sampling and hard rasterization (e.g., `faces_per_pixel=1`); the rendered foreground is composited over the CARLA background with $M_{\mathrm{fg}}$. Under EOT, we randomize camera pose $\phi_{\mathrm{cam}}$ (azimuth/elevation/distance) and environment $e$ (weather/time, map); environment variation enters via the CARLA background and Intrinsic Appearance Transfer (IAT), not the forward renderer, and camera intrinsics remain fixed unless noted.

## 3.3 INTRINSIC APPEARANCE TRANSFER (IAT)

Prior pipelines attach a learned "environment module" to inject illumination and weather before compositing (e.g., EFE/NRP in RAUCA, DTA/ACTIVE, diffusion-guided CNCA) (Zhou et al., 2025; Suryanto et al., 2022; 2023; Lyu et al., 2024). Reimplementations in this style (using render–vehicle pairs from CARLA) generalized poorly to *unseen* colors/textures when evaluated under matched viewpoint/environment. We therefore replace heavy encoder–decoders with a fully differentiable radiometric model that carries better out of distribution.

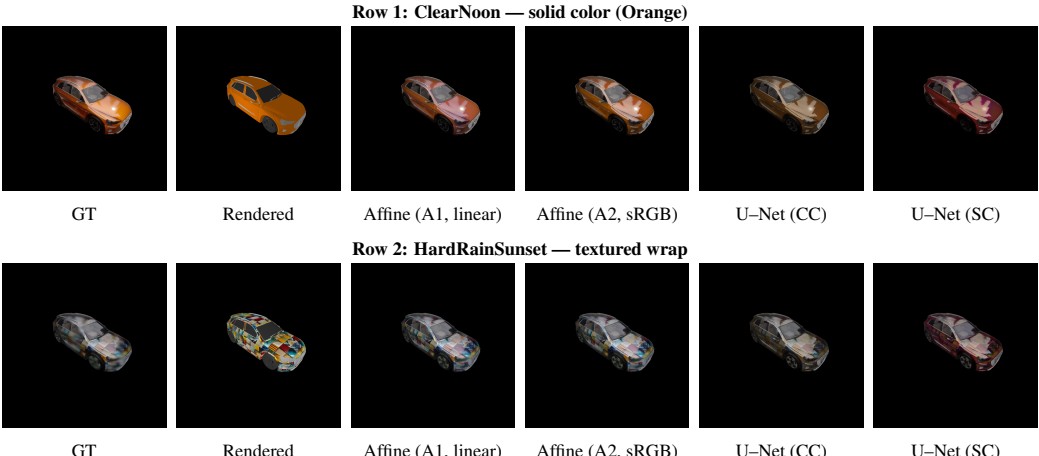

Figure 2: **IAT qualitative comparison** on a solid color (*Orange*) and a held-out texture. Columns: ground truth, bare renderer, best closed-form affine in linear RGB (A1), best closed-form affine in sRGB (A2), best U-Net with cross-color training (CC), and best U-Net with same-color training (SC). All selections follow vehicle-masked metrics.

Given $I_r$ and $I_{\text{veh}}$, IAT predicts per-pixel, per-channel multiplicative/additive carriers $(A, V)$ and outputs $\tilde{I} = A \odot I_r + V$, with gradients flowing through the renderer, UV sampling, and IAT into the StyleGAN latent.

We consider two realizations (both differentiable). *Closed-form affine (default):* $(A, V)$ are estimated at each pixel/channel by robust, regularized regression over *cross-reflectance* samples taken at the same pose $\phi$ and environment $e$ but different base paints/textures. For samples $\{(x_s, y_s)\}_s \equiv \{(I_r^{(s)}, I_{\text{veh}}^{(s)})\}$,

$$\min_{a, v} \sum_s w_s \, \rho_\delta(a \, x_s + v - y_s) + \lambda(a^2 + v^2),$$

where $\rho_\delta$ is the Huber loss, $w_s$ are mask-normalized inverse-area weights (equalizing small/large vehicles), and $\lambda$ is an $\ell_2$ shrinkage. Practical guards—minimum support, gain floors, coefficient clipping, optional robust reweighting, and an optional local low-pass—stabilize estimates; fits can be performed in linear RGB or sRGB. *U-Net carrier predictor (ablation):* a compact U-Net $h_\theta$ takes a masked reference vehicle crop and predicts six channels, passed through a sigmoid to yield $(A, V) \in (0, 1)^{H \times W \times 3}$, after which $\tilde{I} = A \odot I_r + V$. Training covers same-color (SC) and cross-color (CC) pairings. For the U-Net variant we use a configurable objective family: masked $\ell_1/\ell_2$ ($\mathcal{L}_{\text{L1-m}}, \mathcal{L}_{\text{MSE-m}}$), unmasked $\ell_1/\ell_2$ ($\mathcal{L}_{\text{L1}}, \mathcal{L}_{\text{MSE}}$), inverse-ratio variants, and perceptual/style terms ($\mathcal{L}_{\text{LPIPS-m}}, \mathcal{L}_{\text{content-m}}, \mathcal{L}_{\text{style-m}}$); SSIM/PSNR are reported for monitoring. Mask normalization and inverse-area weighting are used throughout to balance object sizes.

Empirically, environment networks in the style of RAUCA/FCA/ACTIVE/DTA/CNCA underperformed on *unseen* colors/textures at matched $(e, \phi)$. In contrast, the closed-form affine IAT—fit once from training colors and applied to held-out colors/textures—achieved consistently lower masked reconstruction error and stronger cross-condition stability, while remaining training-free at test time and fully differentiable for end-to-end attacks.

### 3.4 GAN-PRIOR CONSTRAINED TEXTURE ATTACK

As in Fig. 1, gradients traverse renderer $\rightarrow$ Intrinsic Appearance Transfer (IAT) $\rightarrow$ early StyleGAN layers. We adopt a pre-trained StyleGAN2-ADA prior $G$ and optimize in $W^+$ at coarse layers only. Let $f$ be the mapping network and $S$ the synthesis network with $L$ style injection points. We sample

$$\mathbf{z}_0 \sim \mathcal{N}(\mathbf{0}, \mathbf{I}), \quad \mathbf{w}_0 = f(\mathbf{z}_0), \quad \mathbf{W}_0 = \underbrace{[\mathbf{w}_0, \ldots, \mathbf{w}_0]}_{L \text{ copies}} \in \mathbb{R}^{L \times d},$$

$$P_0 = S(\mathbf{W}_0; \boldsymbol{\xi}_0) \in [0, 1]^{256 \times 256 \times 3}, \quad T = P_0.$$

Table 1: **IAT quality (masked, vehicle-only)** on *unseen color* (C) and *unseen textures* (T). Bold = best *within block* (U-Net vs. Affine). Underline = best *overall*. We report PSNR, SSIM, LPIPS, plus masked $L_1$ and masked MSE. Affine symbols: (A1) Linear RGB; Huber+$\ell_2$. (A1$^{\mathrm{LP}}$) Linear RGB; Huber+$\ell_2$ with $3{\times}3$ low-pass. (A2) sRGB; Huber+$\ell_2$. (A0) Linear RGB; simple $\ell_2$. (A0s) sRGB; simple $\ell_2$. U-Net rows use CC=C2OC and SC=C2SC with masked losses (L1/MSE/CONTENT).

| Method | Setup | Unseen color (C) | | | | | Unseen textures (T) | | | | |
|---|---|---|---|---|---|---|---|---|---|---|---|
| | | PSNR↑ | SSIM↑ | LPIPS↓ | $L_1 \downarrow$ | MSE↓ | PSNR↑ | SSIM↑ | LPIPS↓ | $L_1 \downarrow$ | MSE↓ |
| U-Net | CC / L1 (masked) | **20.650** | 0.725 | 0.247 | **0.068** | **0.011** | 20.559 | 0.636 | 0.346 | 0.070 | 0.011 |
| U-Net | CC / MSE (masked) | 20.508 | 0.704 | 0.261 | 0.072 | **0.011** | 20.614 | 0.636 | 0.292 | 0.071 | 0.010 |
| U-Net | CC / CONTENT (masked) | 20.564 | **0.753** | **0.193** | 0.071 | **0.011** | **21.108** | 0.661 | **0.235** | **0.067** | **0.010** |
| U-Net | SC / L1 (masked) | 18.441 | 0.699 | 0.327 | 0.085 | 0.017 | 19.184 | **0.671** | 0.327 | 0.078 | 0.014 |
| U-Net | SC / MSE (masked) | 18.313 | 0.673 | 0.334 | 0.088 | 0.017 | 19.001 | 0.642 | 0.330 | 0.081 | 0.014 |
| U-Net | SC / CONTENT (masked) | 18.142 | 0.685 | 0.298 | 0.090 | 0.018 | 18.701 | 0.660 | 0.311 | 0.085 | 0.015 |
| A1 | Closed-form | 25.948 | 0.838 | 0.137 | 0.039 | 0.004 | **24.829** | **0.736** | 0.214 | **0.040** | **0.004** |
| A1$^{\mathrm{LP}}$ | Closed-form | 24.410 | 0.759 | 0.233 | 0.049 | 0.005 | 23.629 | 0.668 | 0.283 | 0.050 | 0.006 |
| A2 | Closed-form | **26.164** | 0.842 | **0.130** | **0.035** | **0.003** | 24.391 | 0.730 | **0.212** | 0.043 | 0.005 |
| A0 | Closed-form | 25.926 | **0.844** | 0.137 | 0.038 | 0.004 | 24.627 | 0.730 | 0.218 | 0.041 | **0.004** |
| A0s | Closed-form | 26.104 | 0.841 | 0.131 | 0.036 | **0.003** | 24.304 | 0.728 | 0.214 | 0.044 | 0.005 |

To bias toward macro-material edits (palette, large-scale structure) that are physically reproducible, we restrict updates to a coarse layer set $\mathcal{L} = \{0, \ldots, k\}$ and freeze finer layers. Writing $\mathbf{W} = \mathbf{W}_0 + \Delta\mathbf{W}$, we enforce the hard constraint $\Delta\mathbf{w}^{(\ell)} = \mathbf{0} \ \forall \ell \notin \mathcal{L}$ via masked updates

$$\mathbf{W} \leftarrow \mathbf{W} - \eta\big(\mathbf{M} \odot \nabla_{\mathbf{W}}\mathcal{L}_{\text{attack}}\big), \qquad \mathbf{M}[\ell] = \Vmathbb{1}[\ell \in \mathcal{L}],$$

followed by gradient clipping. Generator weights and noise remain frozen; the synthesized UV is re-computed each step, $P = S(\mathbf{W}; \boldsymbol{\xi}_0)$, and passed to the renderer. We regularize the trainable styles toward the pre-trained average style $\bar{\mathbf{w}}$ via

$$\mathcal{R}_w = \tfrac{1}{|\mathcal{L}|} \sum_{\ell \in \mathcal{L}} \big\| \mathbf{w}^{(\ell)} - \bar{\mathbf{w}} \big\|_2^2,$$

weighted by $\lambda_w$ (see Sec. 3.5). Pixel-space TV is disabled by default because early-layer latent updates already produce smooth, band-limited textures; we only ablate TV or Fourier penalties when explicitly stated.

## 3.5 LOSSES AND OPTIMIZATION

Our detection-side objective follows the standard structure used in full-coverage attacks (e.g., FCA, ACTIVE, RAUCA): suppress objectness, confuse classification, and degrade localization under EOT. Our distinct (but not novel) choices are an *entropy-based* classification term for untargeted confusion and an explicit *GIoU* penalty for localization, which we found to yield stable gradients in our StyleGAN–IAT pipeline.

We optimize against a white-box detector $\mathcal{D}$ (YOLOv3 in our main experiments) and evaluate black-box transfer to YOLOv8, Faster R-CNN, and Deformable DETR. For a frame $I$ rendered under an EOT sample $\phi = (\phi_{\text{cam}}, e)$, the detector emits, per anchor/cell $i$, an objectness score $p_i^{\text{obj}} \in [0, 1]$, class probabilities $\mathbf{q}_i \in \Delta^{C-1}$, and a box $\mathbf{b}_i$. Let $H(\mathbf{q}) = -\sum_c q(c) \log q(c)$ denote entropy. With ground-truth $\mathbf{b}_i^\star$ from CARLA (assigned by the detector's matching rule), we define

$$\mathcal{L}_{\text{obj}} = \frac{1}{N} \sum_{i=1}^{N} \text{BCE}(p_i^{\text{obj}}, 0), \quad \mathcal{L}_{\text{cls}}^{\text{unt}} = -\frac{1}{N} \sum_{i=1}^{N} H(\mathbf{q}_i),$$

$$\mathcal{L}_{\text{cls}}^{\text{tgt}} = \frac{1}{N} \sum_{i=1}^{N} \big[ -\log q_i(c_t) \big], \quad \mathcal{L}_{\text{loc}} = \frac{1}{N} \sum_{i=1}^{N} \big(1 - \text{GIoU}(\mathbf{b}_i, \mathbf{b}_i^\star)\big).$$

We aggregate across scales/anchors and EOT:

$$\mathcal{L}_{\text{attack}} = \lambda_{\text{obj}} \, \mathbb{E}_{\phi \sim \mathcal{T}}[\mathcal{L}_{\text{obj}}] + \lambda_{\text{cls}} \, \mathbb{E}_{\phi \sim \mathcal{T}}[\mathcal{L}_{\text{cls}}] + \lambda_{\text{loc}} \, \mathbb{E}_{\phi \sim \mathcal{T}}[\mathcal{L}_{\text{loc}}].$$

The latent regularizer acts only on trainable styles (Sec. 3.4): $\mathcal{R}_w = \frac{1}{|\mathcal{L}|} \sum_{\ell \in \mathcal{L}} \| \mathbf{w}^{(\ell)} - \bar{\mathbf{w}} \|_2^2$.

Table 2: Comparative performance of adversarial camouflage methods against five object detectors. AP@0.5 values (lower is better) are reported; parentheses show percentage change relative to **No Camouflage**.

| Method | DINO | FRRCN | RetinaNet | RTMDet | YOLOv3 |
|---|---|---|---|---|---|
| **No Camouflage** | 0.86 (0%) | 0.72 (0%) | 0.72 (0%) | 0.79 (0%) | 0.75 (0%) |
| DAS Wang et al. (2021) | 0.85 (-1.44%) | 0.59 (-17.84%) | 0.67 (-6.02%) | 0.72 (-8.17%) | 0.69 (-7.48%) |
| FCA Wang et al. (2022) | 0.81 (-5.58%) | 0.37 (-47.94%) | 0.50 (-29.90%) | 0.54 (-30.83%) | 0.33 (-55.50%) |
| RAUCA Zhou et al. (2024) | 0.64 (-25.29%) | 0.13 (-82.20%) | 0.24 (-65.86%) | 0.26 (-66.34%) | 0.11 (-84.61%) |
| **Ours** | **0.59 (-31.73%)** | **0.13 (-82.46%)** | **0.22 (-68.66%)** | **0.26 (-67.12%)** | **0.11 (-85.78%)** |

The overall attack objective under EOT is

$$\min_{\mathbf{W}} \ \mathbb{E}_{\phi \sim \mathcal{T}} \big[ \mathcal{L}_{\text{attack}}(\mathcal{D}(I(\mathbf{W}, \phi))) \big] + \lambda_w \, \mathcal{R}_w \quad \text{s.t.} \quad \frac{\partial \mathcal{L}}{\partial \mathbf{w}^{(\ell)}} = 0 \ \forall \, \ell \notin \mathcal{L},$$

with $I$ produced by renderer $\rightarrow$ Intrinsic Appearance Transfer (IAT) $\rightarrow$ composite (Sec. 3.1). We use Adam with masked gradients for the layer cap, mini-batch EOT sampling over $\phi$, and standard gradient clipping. Pixel-TV on $P$ is *not* used by default; anti-aliasing is handled in rendering (mipmaps/area sampling). The IAT module is trained offline with masked pixel/perceptual/style losses and cross-reflectance supervision.

## 4 EXPERIMENTS

### 4.1 EXPERIMENTAL SETUP

**Dataset.** We render a grid over weather/time presets ($|\mathcal{W}| = 22$: *Clear/Cloudy* $\times$ {Noon, Sunset, Night}, *Wet* $\times$ {Noon, Sunset, Night}, *WetCloudy* $\times$ {Noon, Sunset, Night}, *SoftRain/HardRain* $\times$ {Noon, Sunset, Night}, *MidRainy* $\times$ {Noon, Sunset, Night}, and *DustStorm*), camera azimuths ($|\mathcal{A}| = 8$: 0:45:315°), elevations ($|\mathcal{E}| = 9$: [5:10:85]°), distances ($|\mathcal{D}| = 6$: {5, 10, 15, 20, 25, 30} m), and map locations ($|\mathcal{L}| = 3$). For each configuration we render one body appearance drawn from 11 uniform colors and 6 textured wraps ($|\mathcal{S}| = 17$). This yields $|\mathcal{W}| \cdot |\mathcal{A}| \cdot |\mathcal{E}| \cdot |\mathcal{D}| \cdot |\mathcal{L}| = 22 \cdot 8 \cdot 9 \cdot 6 \cdot 3 = 28{,}512$ unique camera–environment configurations and $28{,}512 \cdot 17 = 484{,}704$ images per vehicle. We form cross-reflectance pairs at fixed $(\phi, e)$ for Intrinsic Appearance Transfer (IAT) and drive EOT during attack optimization. *IAT split:* we train Intrinsic Appearance Transfer (IAT) on **8** uniform colors; the remaining **3** colors and all **6** textured wraps are held out and used only for evaluation.

**Baseline methods.** We benchmark against three full-coverage, renderer-driven pipelines under CARLA settings comparable to ours: **DAS** (Wang et al., 2021), **FCA** (Wang et al., 2022), and **RAUCA** (Zhou et al., 2024; 2025). In addition, to isolate the contribution of our appearance-transfer module, we reimplement a **U-Net environment-transfer** baseline ("U-Net EFE") in the RAUCA style but trained with our cross-reflectance recipe and masked, area-normalized losses (L1/MSE/Content/Style/LPIPS variants; see Sec. 3.3). All methods use the same EOT distributions, camera sampling, weather presets, UV atlas, and detector protocols for fair comparison.

**Target detectors.** For strict comparability with prior full-coverage pipelines, we perform *white-box* optimization only on **YOLOv3** (COCO-pretrained). *Black-box* transfer is evaluated—without any test-time finetuning—on **Faster R-CNN** (Ren, 2015), **RetinaNet** (Lin, 2017), **RTMDet** (Lyu et al., 2022), and **DINO** (Zhang et al., 2022). All detectors use the same preprocessing pipeline (resize/letterbox), single-scale inference, and identical confidence/NMS thresholds as the white-box model; we report COCO mAP@[.5:.95] and $\Delta$ mAP for all methods.

**Training details.** Unless noted, we jointly optimize the StyleGAN latent (early layers only) and Intrinsic Appearance Transfer (IAT) with Adam; EOT samples $K=2$ views per step over camera elevation/azimuth/distance and environment $e$. We use a latent-norm prior and a UV band-limit. Tiling $\tau$ is set to 1 by default and appears only in ablations.

### 4.2 IAT EVALUATION (STANDALONE)

Table 1 summarizes Intrinsic Appearance Transfer (IAT) fidelity on *unseen colors* (C) and *unseen textures* (T) using vehicle-masked metrics, and Fig. 2 shows representative qualitative results. Within

Table 3: **IAT qualitative comparison** on a solid color (*Orange*, ClearNoon) and a held-out texture (HardRainSunset). Columns: ground truth, bare renderer, best affine in linear RGB (A1), best affine in sRGB (A2), best U-Net (CC), and best U-Net (SC).

| Method | DINO | FRRCN | RetinaNet | RTMDet | YOLOv3 |
|---|---|---|---|---|---|
| **No Camouflage** | 0.86 (0%) | 0.72 (0%) | 0.72 (0%) | 0.79 (0%) | 0.75 (0%) |
| DAS Wang et al. (2021) | 0.86 (+0.00%) | 0.63 ( -12.50%) | 0.69 ( -4.17%) | 0.74 ( -6.33%) | 0.71 ( -5.33%) |
| FCA Wang et al. (2022) | 0.83 ( -3.49%) | 0.45 ( -37.50%) | 0.55 ( -23.61%) | 0.60 ( -24.05%) | 0.40 ( -46.67%) |
| RAUCA Zhou et al. (2024) | 0.68 ( -20.93%) | 0.20 ( -72.22%) | 0.30 ( -58.33%) | 0.33 ( -58.23%) | 0.16 ( -78.67%) |
| **Ours** | **0.61 ( -29.07%)** | **0.15 ( -79.17%)** | **0.24 ( -66.67%)** | **0.28 ( -64.56%)** | **0.12 ( -84.00%)** |

the U-Net family, *cross-color* (CC) training consistently generalizes better than *same-color* (SC): CC/**CONTENT** achieves the strongest *perceptual/structural* match (best SSIM/LPIPS) on both C and T, and it also gives the lowest masked $L_1$ and MSE on T. CC/$L_1$ slightly edges CC/CONTENT on C for *distortion* metrics (PSNR, $L_1$, MSE), but trails on SSIM/LPIPS; CC/**MSE** is competitive on MSE yet weaker perceptually. In short, if the goal is *generalization to textured wraps* and human-perceived fidelity, CC/CONTENT is our preferred U-Net training recipe; if one exclusively optimizes radiometric error on solid colors, CC/$L_1$ offers a small PSNR/$L_1$ advantage.

For closed-form *affine* variants, trends are complementary. The sRGB-domain Huber+$\ell_2$ fit (**A2**) is best on *unseen colors* (highest PSNR, lowest $L_1$/MSE, best LPIPS), suggesting that estimating multiplicative/additive carriers in the gamma-encoded space preserves color/tonal relationships most faithfully for uniform paints. Conversely, the linear-RGB Huber+$\ell_2$ fit (**A1**) yields the strongest *unseen texture* performance (best PSNR/SSIM/$L_1$/MSE, with LPIPS essentially tied), indicating that operating in a linear radiometric space better handles illumination–reflectance interactions once high-frequency texture is present. Simple $\ell_2$ baselines (A0/A0s) remain close but slightly behind, while adding a $3\times3$ low-pass (A1$^{\text{LP}}$) harms detail (LPIPS/SSIM). Taken together, we adopt **A2** as the default (more color-accurate and best overall on C), and note that **A1** is a strong drop-in when texture generalization or SSIM/PSNR on wraps is prioritized.

Finally, the metric suite is complementary: PSNR/$L_1$/MSE quantify radiometric fidelity inside the vehicle mask, SSIM emphasizes structural luminance/contrast, and LPIPS measures perceptual similarity in deep feature space. Cross-color supervision (CC) forces color-invariant, illumination-consistent carriers, explaining the observed robustness on textured wraps; SC variants, trained on identical colors, overfit chroma/tonal statistics and degrade under color/texture shift.

Table 4: Average AP@0.5 across five detection models (from Table 2). Lower is better.

| Camouflage | Average AP@0.5 |
|---|---|
| no_camouflage | 0.7538 |
| DAS Wang et al. (2021) | 0.6825 |
| FCA Wang et al. (2022) | 0.4788 |
| RAUCA Zhou et al. (2024) | 0.2975 |
| **Ours** | **0.2863** |

## 4.3 CAMOUFLAGE ATTACK EVALUATION

**Main results (no IAT at evaluation).** Table 2 reports AP@0.5 (lower is better) across five detector families. Relative to the *No Camouflage* baseline (mean 0.768 across detectors), our method yields the lowest average AP (0.262), a 66% reduction, outperforming DAS Wang et al. (2021) (avg 0.704; 8% reduction), FCA Wang et al. (2022) (avg 0.510; 34% reduction), and RAUCA Zhou et al. (2024) (avg 0.276; 64% reduction). Per detector, we reduce AP from $0.75 \rightarrow 0.11$ on YOLOv3 ($-85.8\%$), $0.72 \rightarrow 0.13$ on Faster R-CNN ($-82.5\%$), $0.72 \rightarrow 0.22$ on RetinaNet ($-68.7\%$), $0.79 \rightarrow 0.26$ on RTMDet ($-67.1\%$), and $0.86 \rightarrow 0.59$ on DINO ($-31.7\%$). Against the strongest prior (RAUCA), we match or improve on all five models, with the largest margins on DINO ($0.64 \rightarrow 0.59$; $-6.4$ points vs. baseline) and RetinaNet ($0.24 \rightarrow 0.22$; $-2.8$ points), and near-parity on YOLOv3/RTMDet.

**With IAT applied at evaluation.** To assess robustness when appearance transfer is present at test time, we apply IAT to all methods (Table 3). As expected, AP values rise slightly for every method, yet ours remains best on *every* detector (e.g., YOLOv3 0.12 vs. RAUCA 0.16; Faster R-CNN 0.15

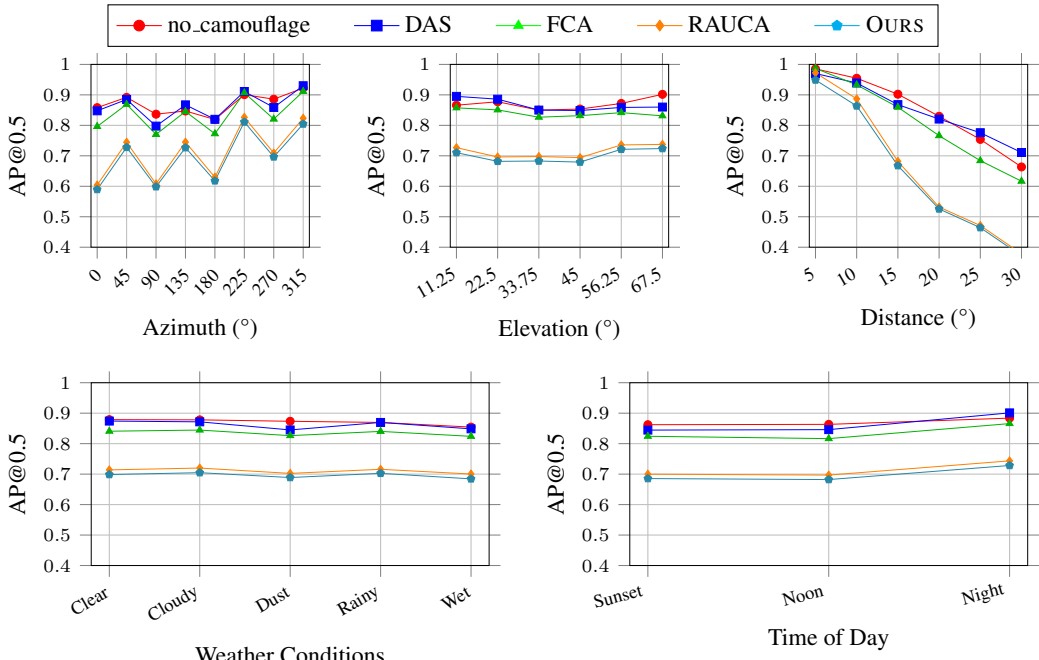

Figure 3: Performance comparison of different camouflage methods across viewing angles, distances, weather conditions, and time of day. Lower AP@0.5 values indicate stronger effectiveness.

vs. 0.20; RetinaNet 0.24 vs. 0.30; RTMDet 0.28 vs. 0.33; DINO 0.61 vs. 0.68). Averaged over detectors, our AP increases modestly from 0.262 to 0.280 (still a 63.6% drop vs. baseline), indicating that the combination of UV-first rendering, EOT over camera+environment, and latent-constrained textures transfers well even when evaluation includes illumination/veiling adjustment.

**Average summary.** Table 4 aggregates AP@0.5 across detectors (lower is better). Our method achieves the best average (0.2863), improving over RAUCA (0.2975), FCA (0.4788), and DAS (0.6825), and far below the no-camouflage baseline (0.7538). These gains reflect both stronger source-model suppression and improved cross-model transfer.

**Sensitivity to viewpoint and environment.** Figure 3 breaks down AP@0.5 across azimuth, elevation, distance, weather, and time of day. Our curve is consistently the lowest (i.e., strongest attack) across bins, with clear margins at longer ranges and under adverse weather. This aligns with our design: (i) a Blender-remapped UV atlas and PyTorch3D renderer that stabilize gradients to texels; (ii) an EOT that emphasizes physically dominant factors (camera pose and environment); and (iii) a StyleGAN prior restricted to coarse layers, yielding material-like textures that avoid overfitting to any single detector or condition.

## 5 CONCLUSION

We presented an environment-aware, on-manifold approach to physical camouflage for vehicle detection. By enforcing illumination consistency with a closed-form affine IAT and constraining optimization to early layers of a StyleGAN prior, the method delivers realistic, printable wraps while achieving strong white-box suppression and black-box transfer across detector families. The end-to-end pipeline (Blender UV remap, PyTorch3D rasterization, camera + environment EOT) consistently lowers AP@0.5 versus prior full-coverage methods, and retains performance even when IAT is applied at evaluation.

**Limitations and future work.** We did not conduct full real-world print trials or evaluate beyond the vehicle class. Future directions include multi-object scenes, broader materials (e.g., metallic/pearlescent finishes), richer sensor/codec models, and controlled physical studies to further close the sim-to-real gap.

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
