# OpenReview forum: "Environment-Aware On-Manifold 3D Texture Camouflage for Physical Attacks on Vehicle Detectors"
_ICLR.cc/2026/Conference — ICLR 2026 Conference Withdrawn Submission_

### Official Review · Reviewer_95Mx · 2025-10-27

**Soundness:** 2
**Presentation:** 1
**Contribution:** 2
**Rating:** 2
**Confidence:** 4

**Summary:**

This paper presents a technical pipeline for designing and implementing full-coverage, printable 3D camouflage patterns that can be physically applied to vehicles in order to reduce or evade detection by camera-based vehicle detectors. The method uses an Intrinsic Appearance Transfer (IAT) module for adapting lighting and environmental conditions to the rendered vehicles and textures generated using a StyleGAN-based model, with adversarial optimization restricted to the early layers of the generator for realism. Experimental evaluations in simulated environments confirm the pipeline's capability to suppress detection scores across several state-of-the-art object detectors under a variety of camera views, weather conditions, and distances.

**Strengths:**

The strengths of this paper are the following:

- The paper presents a novel Intrinsic Appearance Transfer (IAT) module that decouples scene photometry from texture optimization, providing robust realism across diverse environmental and lighting conditions.
- The proposed pipeline leverages StyleGAN-based UV texture priors and restricts updates to physically meaningful layers, resulting in printable, realistic camouflage textures that maintain attack efficacy and transfer across models.
- Extensive testing across weather, time, pose, and location settings confirms  attack performance and robustness, demonstrating cross-condition and cross-model effectiveness for vehicle detector attacks.

**Weaknesses:**

- **Limited originality:**

    The approach is very similar to previous works like DTA, CNCA, and RAUCA. It mainly combines existing methods pipelines with a generative model (StyleGAN), so I think the knowldege contribution  is not very unique.

- **Lack of real-world testing:**

    All results are from simulations, and there are no actual physical experiments using printed camouflage. This means the method is not proven to work in real-life scenarios. They didn't compare their method with the similar work CNCA.

- **Weaknesses in writing and explanation:**

    The paper does not clearly explain the background, motivation, or the limitations of past research. The abstract and introduction are hard to follow, and the paper could be improved by making the writing more logical and storytelling.

    **Reference:**

    Suryanto, N., Kim, Y., Kang, H., Larasati, H. T., Yun, Y., Le, T. T. H., ... & Kim, H. (2022). Dta: Physical camouflage attacks using differentiable transformation network. In *Proceedings of the IEEE/CVF Conference on Computer Vision and Pattern Recognition* (pp. 15305-15314).

    Zhou, J., Lyu, L., He, D., & Li, Y. (2024). Rauca: A novel physical adversarial attack on vehicle detectors via robust and accurate camouflage generation. *arXiv preprint arXiv:2402.15853*.

    Lyu, L., Zhou, J., He, D., & Li, Y. (2024). CNCA: Toward Customizable and Natural Generation of Adversarial Camouflage for Vehicle Detectors. *arXiv preprint arXiv:2409.17963*.

**Questions:**

My primary concern regarding this paper is that its knowledge and contribution are not clearly presented. I would appreciate it if the author could clarify the true contribution of this work and help highlight what is novel and significant about their research.  If the author can provide more explanation. I will reconsider my rating accordingly.

**Details Of Ethics Concerns:**

This work is about adversarial attacks against vehicle detection systems. These attacks could potentially be exploited by malicious users, which raises significant ethical concerns. I recommend that the authors address these ethical implications and discuss how their work considers or mitigates risks associated with potential misuse.

---

### Official Review · Reviewer_Bar2 · 2025-10-30

**Soundness:** 2
**Presentation:** 3
**Contribution:** 2
**Rating:** 4
**Confidence:** 3

**Summary:**

This work presents a novel technique for generating realistic, environment-aware 3D adversarial camouflage. The core of the method is a closed-form Intrinsic Appearance Transfer (IAT) module that models illumination and weather effects without training, coupled with a StyleGAN framework to produce physically printable textures. When tested in the CARLA simulator across various poses, times, and weather scenarios, the camouflage proved highly effective, slashing the detection AP of systems like YOLOv3 (−85.8%) and Faster R-CNN (−82.5%) and surpassing the performance of prior attacks.

**Strengths:**

+ The method integrates IAT for illumination modeling with StyleGAN latent optimization to effectively balance realism and attack efficacy.

+ Based on the Carla dataset, the authors evaluate the robustness across different kinds of detectors under different poses.

**Weaknesses:**

+ No real-world validation: Although the authors mention this in the limitations, the key contribution of this work is a printable 3D camouflage attack method. However, the experiments are mainly conducted in simulation environments.

+ The proposed method is only tested on the outdoor environment and the vehicle category.

+ Unclear robustness to domain shift (StyleGAN prior dependency) or runtime practicality.

**Questions:**

1. Have you actually printed and tested any camouflage in real-world lighting conditions? If not, what evidence suggests the IAT module will generalize beyond simulation (e.g., sensor noise, material reflectance differences)?
2. Why does DINO remain relatively robust compared to other conv-based methods?
3. Is the proposed method robust to different view angle?

---

### Official Review · Reviewer_mFsC · 2025-11-01

**Soundness:** 1
**Presentation:** 1
**Contribution:** 2
**Rating:** 0
**Confidence:** 5

**Summary:**

This paper presents an environment-aware physical adversarial camouflage framework intended to evade vehicle detectors. The framework is very similar to RAUCA [1], but introduces two main modifications:

1. Replacing RAUCA’s NRP (a U-Net environment feature extractor) with a closed-form Intrinsic Appearance Transfer (IAT) module that performs per-pixel affine illumination transfer.
2. Constraining adversarial texture optimization using a StyleGAN latent prior, restricting updates to early layers to encourage texture naturalness and smoothness.

The authors compare IAT against a learned NRP/U-Net under different training regimes and loss functions, and evaluate transfer to three held-out colors and six textured wraps using rendered ground truth from CARLA. They report that different IAT and training-loss combinations yield different optimal results (sRGB affine performs best on unseen colors, while linear-RGB affine works better on unseen textures).

Adversarial camouflages are evaluated only in CARLA simulation datasets across weather, time, and camera pose variations, and compared against prior baselines (DAS [2], FCA [3], RAUCA [1]), showing only marginal improvements over the best existing methods. The reported gains are also limited to digital AP reductions with no physical-world evaluation is provided.

Overall, the contribution is incremental, consisting of a simplified affine illumination transfer and an on-manifold StyleGAN constraint. The experimental setup is insufficient to substantiate the paper’s claims of broad generalization or physical robustness. The presentation quality is also weak, with errors in some figures and tables, underexplained hyperparameters, and missing details, making the manuscript not ready for publication. See the Weaknesses section for detailed points.

---

References:
1. Jiawei Zhou, Linye Lyu, Daojing He, and Yu Li. Rauca: a novel physical adversarial attack on vehicle detectors via robust and accurate camouflage generation. In Proceedings of the 41st International Conference on Machine Learning, pp. 62076–62087, 2024.
2. Jiakai Wang, Aishan Liu, Zixin Yin, Shunchang Liu, Shiyu Tang, and Xianglong Liu. Dual attention suppression attack: Generate adversarial camouflage in physical world. In Proceedings of the IEEE/CVF conference on computer vision and pattern recognition, pp. 8565–8574, 2021.
3. Donghua Wang, Tingsong Jiang, Jialiang Sun, Weien Zhou, Zhiqiang Gong, Xiaoya Zhang, Wen Yao, and Xiaoqian Chen. Fca: Learning a 3d full-coverage vehicle camouflage for multi-view physical adversarial attack. In Proceedings of the AAAI conference on artificial intelligence, volume 36, pp. 2414–2422, 2022.

**Strengths:**

1. The proposed IAT module provides a simpler and computationally efficient differentiable formulation for environment feature transfer compared to prior neural network based modules such as RAUCA’s NRP, assuming its claimed behavior is well validated.
2. The paper reports consistent improvements in both the quality of environment transfer (IAT compared with U-Net baselines) and the overall adversarial camouflage effectiveness (mAP reductions) across several detectors, even though the gains are modest.

**Weaknesses:**

The paper has major weaknesses in both **soundness** and **presentation**, making it not ready for publication.

**Soundness**

1. **Limited novelty.**
   The contribution is incremental. The IAT is essentially a simplified affine illumination model replacing RAUCA’s U-Net environment module, and the use of a StyleGAN latent prior for texture control is a straightforward adaptation rather than a substantive methodological advance.

2. **Insufficient validation of IAT generalization.**
   The IAT evaluation uses only **three held-out colors and six textured wraps**, which is far too small to justify claims of generalization. The reported observation that different losses or color spaces yield “optimal” results for different cases is statistically weak and may be misleading given the small and unbalanced sample size.

3. **Inadequate evaluation of adversarial camouflage.**
   Although the paper emphasizes “environment-aware” and “physically printable” camouflage, all results are obtained within their **own differentiable rendering pipeline**, not on fully rendered CARLA scenes or any real or physically simulated setup. CARLA is used only to render reference images and environmental backgrounds, not for testing attack effectiveness in realistic scenes. The method is therefore evaluated only on **digitally composited images**, which does not validate its claimed physical robustness. Moreover, important baselines such as **DTA**, **ACTIVE**, and **CNCA** are mentioned but not included in comparison. There is also no real-world comparison, which is a critical setting for physical attacks.

4. **Incomplete ablation of StyleGAN effects.**
   The StyleGAN latent prior is highlighted as a main contribution, but no ablation is provided to isolate its influence on texture realism, smoothness, or attack strength. Without such analysis, the claimed benefits of the StyleGAN component remain unsubstantiated.

5. **Lack of qualitative visualization.**
   The paper does not provide qualitative examples of the final adversarial textures or corresponding detection outputs. Readers cannot assess what the generated camouflage looks like or whether it is physically plausible and visually natural.

**Presentation**

1. **Possible LLM-generated or machine-translated phrasing.**
   The manuscript overuses long sentences and em dashes, resulting in unnatural writing and inconsistent formatting.

2. **Errors in Figure 1 and equations.**
   In Figure 1, $B_{\mathrm{bg}}$ (background) is incorrectly shown as an input to the IAT module, while the equation defines $\tilde{I} = \mathrm{IAT}(I_r; I_{\mathrm{veh}}) = A \odot I_r + V$, which is confusing. The figure also shows the output $I$ feeding back into $B_{\mathrm{bg}}$, which is inconsistent with the compositing formula. In addition, the figure does not clearly indicate where the optimized texture is produced.

3. **Error and mismatch in Table 3.**
   Table 3’s caption describes “IAT qualitative comparison on a solid color (Orange, ClearNoon) and a held-out texture (HardRainSunset),” but the table content reports quantitative AP@0.5 results for camouflage performance. This mismatch between caption and content is a serious editorial flaw.

4. **Poor reproducibility.**
   Critical implementation details are missing. Key hyperparameters (for example, the latent regularization weight $\lambda_w$, learning rate, StyleGAN layer configuration, etc.) are not specified. There is no supplementary material detailing dataset splits, textured wraps, or UV maps. As written, the experiments cannot be reliably reproduced.

**Questions:**

Please refer to the Weaknesses section and clarify any misunderstandings or provide additional explanations where applicable.

---

### Official Review · Reviewer_VapM · 2025-11-04

**Soundness:** 2
**Presentation:** 2
**Contribution:** 2
**Rating:** 4
**Confidence:** 3

**Summary:**

This paper investigates a fully-covering and printable 3D camouflage attack against vehicle detectors. The authors replace the commonly used U-Net–based environmental feature extraction approach with a closed-form Intrinsic Appearance Transfer (IAT) module, which demonstrates better qualitative and quantitative performance. In addition, the optimization is constrained by an on-manifold StyleGAN texture prior, aiming to ensure the generated textures remain printable. Under neural-rendered coating simulations, their approach achieves a slight improvement in attack effectiveness compared to previous work.

**Strengths:**

1. The closed-form Intrinsic Appearance Transfer (IAT) module replaces the U-Net–based environmental feature extraction commonly used in prior work, leading to better qualitative and quantitative results.
2. The optimization leverages an on-manifold StyleGAN texture prior to improve the printability of the generated textures.

**Weaknesses:**

1. The main diagram appears problematic. The input to IAT is not the background, and the arrow direction should be Background → I rather than I → Background.

2. There are issues with the titles of Table 3.

3. The paper lacks experiments or ablation studies demonstrating the impact of the introduced StyleGAN prior and the associated loss components.

4. The paper does not provide comparative examples illustrating successful and unsuccessful attacks across different baselines.

**Questions:**

1.Will the authors release the code and the final generated camouflage textures?

---

### Note · Authors · 2025-11-13

I have read and agree with the venue's withdrawal policy on behalf of myself and my co-authors.